Pelagic larval duration, growth rate, and population genetic structure of the tidepool snake moray Uropterygius micropterus around the southern Ryukyu Islands, Taiwan, and the central Philippines

Huang Wen-Chien 1 2
Chang Jui-Tsung 3
Liao Chun 3
Tawa Atsushi 4
Iizuka Yoshiyuki 5
Liao Te-Yu swp0117@gmail.com 2
Shiao Jen-Chieh jcshiao@ntu.edu.tw 1
1 Institute of Oceanography, National Taiwan University , Taipei , Taiwan
2 Department of Oceanography, National Sun Yat-Sen University , Kaohsiung , Taiwan
3 Institute of Bioinformatics and Structural Biology, National Tsing Hua University , Hsinchu , Taiwan
4 National Research Institute of Far Seas Fisheries, Japan Fisheries Research and Education Agency , Orido, Shimizu, Shizuoka , Japan
5 Institute of Earth Sciences, Academia Sinica , Taipei , Taiwan
Robertson D. Ross
Electronic publication date: 2018 May 9
Publication date: 2018
Volume: 6
Electronic Location ID: e4741
Received 2018 Feb 23; Accepted 2018 Apr 19
Copyright: ©2018 Huang et al.
Copyright year: 2018
Copyright holder: Huang et al.
License: This is an open access article distributed under the terms of the Creative Commons Attribution License, which permits unrestricted use, distribution, reproduction and adaptation in any medium and for any purpose provided that it is properly attributed. For attribution, the original author(s), title, publication source (PeerJ) and either DOI or URL of the article must be cited.
License URL: https://creativecommons.org/licenses/by/4.0/

Keywords: Muraenidae, Otolith microstructure, Pelagic larval duration, Population genetic structure

Funding: The authors received no funding for this work.

==============================
The relationships between pelagic larval duration (PLD) and geographic distribution patterns or population genetic structures of fishes remain obscure and highly variable among species. To further understand the early life history of the tidepool snake moray Uropterygius micropterus and the potential relationship between PLD and population genetic structure of this species, otolith microstructure and population genetics based on concatenated mtDNA sequence (cytochrome b and cytochrome oxidase subunit I, 1,336 bp) were analyzed for 195 specimens collected from eight locations around the southern Ryukyu Islands, Taiwan, and the central Philippines. Eels with longer PLD and lower otolith growth rates were observed at relatively higher latitudes with lower water temperatures (54.6 ± 7.7 days and 1.28 ± 0.16 µm day−1 on Ishigaki Island, Japan, vs. 43.9 ± 4.9 days and 1.60 ± 0.19 µm day−1 on Badian, the Philippines), suggesting that leptocephali grew faster and had shortened pelagic periods in warmer waters. Meanwhile, the eels along the southwest coast of Taiwan had relatively longer PLD (57.9 ± 10.5 days), which might be associated with the more complex ocean current systems compared to their counterparts collected along the east coast of Taiwan (52.6 ± 8.0 days). However, the southwestern and eastern Taiwan groups had similar otolith growth rates (1.33 ± 0.19 µm day−1 vs. 1.36 ± 0.16 µm day−1). Despite the intergroup variation in PLD, genetic analysis revealed fluent gene flow among the tidepool snake morays in the study regions, implying that intraspecies PLD variation had a weak effect on genetic structure. The leptocephalus stage might have ensured the widespread gene flow among the study areas and leptocephalus growth was likely influenced by regional water temperature.

Introduction

The population structure of fishes with pelagic larvae is influenced by biological and environmental factors (Leis et al., 2013; Nanninga et al., 2014). However, many of these factors are difficult to parameterize (Nanninga et al., 2014) and pelagic larval duration (PLD) is used as a direct predictor of dispersal potential and population connectivity, especially for site-attached coral reef fishes that do not display migratory behaviors during their juvenile and adult stages (Bowen et al., 2006; Macpherson & Raventos, 2006). Although PLD has been found to strongly influence population genetic structure only in extreme cases with very short or very long PLD (Thresher, Colin & Bell, 1989; Bowen et al., 2001; Weersing & Toonen, 2009), some studies have still suggested that PLD may be a strong determinant for evaluating larval dispersal and population connectivity (e.g., Faurby & Barber, 2012). Variation in PLD may be affected by numerous factors at inter- and intraspecific levels, including genotypes (Tsukamoto, Aoyama & Miller, 2002), physiological conditions of larvae (Reveillac et al., 2008; Han et al., 2010) and environmental changes (Searcy & Sponaugle, 2000; Sponaugle & Pinkard, 2004; Bergenius et al., 2005). Therefore, larval fishes that experience different environmental conditions could have varied early life history traits, leading to specific PLDs among populations (Bay et al., 2006). However, the effects of variable PLDs on population genetics remain unclear and infrequently evaluated for many fish.

Most true eels (Anguilliformes) are demersal fish with limited migration during the juvenile to adult stages (Bassett & Montgomery, 2011; Correia et al., 2012), except for temperate anguillids and some congrids that have offshore spawning areas (Tsukamoto, 2006; Kurogi et al., 2012). The long pelagic larval stage of leptocephalus may play an important role in their distribution and population genetic structures (Miller & McCleave, 2007; Kuroki et al., 2009; Reece et al., 2011). Moreover, variations in the intraspecific PLD of anguillids have been observed within and among geographic regions without genetic divergence. These variations in PLD are likely influenced by nutrition status among individuals (Reveillac et al., 2008; Han et al., 2010). Few studies have been conducted on intraspecific variation in PLDs in marine eels. Kimura et al. (2004) found various PLDs and growth rates for Conger myriaster leptocephali along the east coast of central Japan. Despite the various PLDs, C. myriaster breed in specific spawning areas (Kurogi et al., 2012), which may counteract the influence of PLD variation on their genetic structure. For other marine eel taxa with near-shore spawning strategies, the correlations between variable PLD and genetic structure among geographic areas have yet to be explored thoroughly.

The family Muraenidae, known as moray eels, is the second largest family after the Ophichthidae in the Anguilliformes, with approximately 200 species in 15 genera and two subfamilies (Smith, 2012). Moray eels are broadly distributed in tropical and temperate oceans. Most muraenids inhabit rocky ledges and coral reefs from the intertidal zone to depths of over 300 m, and some species are occasionally found in sandy or freshwater habitats (Tsukamoto et al., 2014). Moray eels have high fidelity to their habitats (Bassett & Montgomery, 2011) and spawn without migrations (Moyer & Zaiser, 1982). Therefore, moray eels are solely dispersed in the pelagic leptocephalus stage, providing an effective means of evaluating the effects of PLD on genetic divergence for marine eels with a local spawning strategy. For example, the tidepool snake moray Uropterygius micropterus (Bleeker, 1852) usually resides in the rocky intertidal zone at depths shallower than 3 m (Chen, 1997), leading to fragmented habitat use across the Indo–Pacific oceans from East Africa to Samoa, north to southern Japan, and south to Australia (Froese & Pauly, 2016). Uropterygius micropterus is a small species measuring less than 40 cm in length (Loh, Shao & Chen, 2011) that has a local spawning strategy. U. micropterus were observed to reproduce in the rocky intertidal zone of Taitung, Taiwan during summer. Numerous males followed and entwined with a female, snapping at the female’s head and trunk. Afterward, a cloud of sperm and transparent, buoyant eggs were discharged into the water (HM Chen, 1994, unpublished data). Due to its small size, local spawning strategy, and specific habitat use, U. micropterus would be a favorable candidate for evaluating larval dispersal and its effects on genetic structure. The present study aimed to (1) examine the otolith microstructure of U. micropterus to evaluate variation in PLD among sampling sites and (2) to test the relationship of differences in PLD to genetic structure.

Materials and Methods

Sample collection

One hundred and ninety-five juvenile and adult U. micropterus were collected by dip net and hand-lining at rocky intertidal zones along Ishigaki Island, Japan (n = 14), six Taiwanese sites (n = 142), and Badian, the Philippines (n = 39) during 2014–2016 (Table 1; Fig. 1). Shitiping (n = 36), Jihui (n = 32), and Green Island (n = 20) are located on the east coast of Taiwan and are influenced by the strong, constant Kuroshio Current that flows northward year-round. Checheng (n = 31), Wanlitong (n = 11), and Liuqiu (n = 12) are located on the southwest coast of Taiwan, in a more complex ocean environment that is affected by numerous water masses on a seasonal basis (Shaw, 1991; Farris & Wimbush, 1996; Hu et al., 2010). Ishigaki is also affected by the Kuroshio Current, and Badian is a relatively closed environment located on Cebu in the central Philippines.

Table 1 Summary of collection data and sample size for the otoliths and mtDNA analysis of U. micropterus used in this study.

The number of alleles, variable sites (S), haplotype diversity (h), and nucleotide diversity (π).

Country	Location	Code	N	N otolith	N mtDNA	No. alleles	S	h	π	
Japan	Ishigaki	IG	14	14	14	14	44	1.0000	0.0063	
Taiwan	Shitiping	ST	36	36	36	31	74	0.9921	0.0060	
	Jihui	JH	32	31	32	31	88	0.9980	0.0073	
	Green Island	GI	20	20	10	10	35	1.0000	0.0067	
	Checheng	CC	31	31	31	30	73	0.9979	0.0061	
	Wanlitong	KT	11	11	11	11	42	1.0000	0.0062	
	Liuqiu	LQ	12	12	12	12	32	1.0000	0.0069	
		Taiwan total	142	141	132	99	157	0.9940	0.0066	
Philippines	Badian	BD	39	39	39	37	85	0.9973	0.0064	
		All specimens	195	194	185	132	179	0.9938	0.0065	

Figure 1 Map of sampling sites for U. micropterus.

Within the pie charts, colors indicate the percentages of pelagic larval duration (PLD) for each sampling site.

Otolith preparation and analysis

Left sagittal otoliths were extracted under stereo microscope, cleaned with deionized water, dried at 55 °C overnight, embedded in Epofix resin, and fixed on microscope slides. The prepared otoliths were grounded along the sagittal plane with 2000 and 2400 grit sandpapers until the core was revealed on the surface, and then polished until smooth with 0.05 µm alumina powder. The ground otoliths were photographed and the growth increments were counted under a compound light microscope (BX-51; Olympus, Tokyo, Japan) from the first feeding check (FFC) to the growth check (GC) (Fig. 2). The otolith growth increments during the leptocephalus stage of U. micropterus were distinguishable and the narrowest rings (approx. 1 µm) were larger than the resolution limitation of the compound light microscope. The GC was defined by Ling, Iizuka & Tzeng (2005) as the prominent check at which the growth increments change from a circular to a radiating pattern, accompanied by a lowered Sr/Ca ratio. This prominent check has been assumed to be associated with the beginning of metamorphosis in leptocephalus and has been used in several studies of marine eel species (Ling, Iizuka & Tzeng, 2005; Lee et al., 2008). Otolith growth increments from the first feeding ring to the GC were expressed as TGC (i.e., PLD). Meanwhile, the radius of GC was measured along the longest axis and divided by TGC to calculate the mean otolith growth rate (µm increment−1, as GGC). Some sub-increments between the wide increments before GC may be ignored, and the blurred rings before FFC were excluded from the count, which may have led to a slight underestimation of the actual number of increments. Fourteen otoliths were randomly chosen for the Sr/Ca ratio analysis to assist in the judgment of GC. Polished otoliths were coated with a layer of carbon and analyzed by an electron probe microanalyzer (EPMA, JXA-8900R; JEOL, Peabody, MA, USA). The Sr/Ca ratios were measured from the core to the edge of each otolith under electron beam conditions of 15 kV and 3 nA, beam size 5 ×4 µm, and 10 µm of spot intervals. Since the otolith growth increments of anguillid species are typically deposited daily in the early leptocephalus and glass eel stages (Sugeha et al., 2001; Shinoda et al., 2004), the otolith increments counted in this study were assumed to be deposited daily (i.e., TGC = days and G GC = µm day−1).

Figure 2 Otolith microstructure.

Otolith microstructure showing the hatch check (HC), first feeding check (FFC), and growth check (GC).

The daily ages and otolith growth rates of U. micropterus at different latitudes were divided into three groups (Ishigaki, Taiwan, and Badian) for statistical analysis. The Taiwanese sampling sites were additionally divided into eastern and southwestern groups, representing different oceanic current conditions at similar latitudes, to test whether ocean currents affected the early life characteristics of U. micropterus despite the small geographic scale. The statistical differences in daily ages and mean daily otolith growth rates between groups were tested by analysis of variance (ANOVA) and the post hoc Tukey HSD test in R (R Development Core Team, 2013). The mean increment widths of sagittal otoliths for intervals of every three rings were also estimated in ImageJ (Abràmoff, Magalhães & Ram, 2004) along the longest axis. The percentages of daily ages for five-day intervals are shown in bar charts.

Population genetic analysis

DNA was extracted from muscle tissue using a Qiagen DNA extraction kit (Qiagen, Hilden, Germany) following the manufacturer’s protocols. Polymerase chain reactions (PCRs) were run in a total volume of 50 µL, including 6 µL of TaKaRa (http://www.clontech.com) 10 × buffer, 4 µL of 2.5 mM dNTPs, 4 µL of 10 µm of each primer, 0.25 µL of TaKaRa Ex Taq DNA polymerase, 6 µL of template DNA at 50 ng/ μL, and 25.75 µL of deionized water. The fragments of cytochrome b (cyt b) 680-bp and cytochrome oxidase subunit I (COI) 656-bp were respectively amplified using the primers cyt b: L14725 (5′-GTG ACT TGA AAA ACC ACC GTT G-3′) (Song, Near & Page, 1998) and H15573 (5′-AAT AGG AAG TAT CAT TCG GGT TTG ATG-3′) (Taberlet, Meyer & Bouvet, 1992); and COI: FishF2 (5′-TCG ACT AAT CAT AAA GAT ATC GGC AC-3′) and FishR2 (5′-ACT TCA GGG TGA CCG AAG AAT CAG AA-3′) (Ward et al., 2005). The annealing temperatures of cyt b and COI were 47 °C and 50 °C, respectively. The thermal profiles of PCR were 94 °C for 5 min, followed by 37 cycles of 94 °C for 1 min, annealing temperature for 45 s, and 72 °C for 1 min, with a final extension at 72 °C for 10 min. The quality of PCR products was checked by electrophoresis with 1.5% agarose gel and then purified using a Macherey-Nagel purification kit (http://www.mn-net.com) according to the manufacturer’s protocols. DNA sequences were generated by an ABI 3730 automated sequencer at the Center for Biotechnology, National Taiwan University. Sequences were assembled and edited manually and aligned using MEGA version 6.0 (Tamura et al., 2013).

Sequences of cyt b and COI were concatenated as a single genetic marker and the analyses that followed were based on this data set. The genetic diversity indexes of haplotype diversity (h) and nucleotide diversity (π) were calculated in DnaSP version 5.0 (Librado & Rozas, 2009) according to Nei (1987). Pairwise ΦST comparisons among sampling sites and among groups with different classes of TGC were estimated in Arlequin version 3.5 (Excoffier & Lischer, 2010), and 10,000 permutations were used to estimate the departure from the null hypothesis of genetic homogeneity. The statistical significance of pairwise ΦST values was adjusted with Bonferroni correction (Rice, 1989) for multiple comparisons. The hierarchical levels of genetic diversity were tested through analysis of molecular variance (AMOVA; Excoffier, Smouse & Quattro, 1992), and the proportions of variations among groups (ΦCT), among populations within groups (ΦSC), and within populations (ΦST) were calculated in Arlequin; 10,000 permutations were used to estimate statistical significance. Two hypothetical grouping treatments were used for AMOVA: (1) based on the three different latitudinal groups of Ishigaki, Taiwan, and Badian; and (2) based on the sampling sites that were associated with three different ocean current conditions, namely the Kuroshio Current system (Ishigaki, Shitiping, Jihui, and Green Island), mixed effect by numerous water masses (Checheng, Wanlitong, and Liuqiu), and the interior current systems of the Philippine archipelago (Badian). The minimum spanning network (MSN) of haplotypes was built using Arlequin version 3.5 and HapStar version 0.7 (Teacher & Griffiths, 2011) to connect haplotypes based on the minimum differences between sequences.

Results

Otolith microstructure and microchemistry

The otolith microstructure of U. micropterus was similar to those of other marine eels (Correia et al., 2004; Ling, Iizuka & Tzeng, 2005; Lee et al., 2008). After polishing, the otolith core became a hole surrounded by a thick ring, referred to as a hatch check (HC; Fig. 2). The first feeding check (FFC) was assumed to form when yolks were absorbed completely and the leptocephali began to ingest external food. There were three to five blurry increments between HC and FFC in some individuals. The increments beyond FFC were circular, and the increment widths gradually increased to a peak of 1.1–2.0 µm at approximately the 10th to 20th increments, followed by a gradual decrease to a minimum of 0.5–0.9 µm. Then, the growth increment width abruptly increased to 1.5–4.0 µm by three to 28 increments and formed a profound growth check (GC). The growth increments after GC were wider (5–15 µm), diffused, and radiative. The Sr/Ca ratios of 14 U. micropterus fluctuated between 3 and 16 ×10−3 from the core to the GC, with no apparent pattern. The Sr/Ca ratios then dropped rapidly, accompanied by the appearance of the GC in all but one individual (Fig. 3).

Figure 3 Patterns of otolith increment widths and Sr/Ca ratios from a Jihui specimen.

Arrows represent the position of the growth check (GC).

Genetic data

One hundred and thirty-two haplotypes from 1,336 bp concatenated mtDNA sequences from 195 U. micropterus were identified (GenBank accession number MF190188–MF190364). In total, 179 polymorphic sites, 112 parsimony informative sites, and 67 singleton variable sites were found. Haplotype diversity (h) and nucleotide diversity (π) ranged from 0.9921 to 1 (average = 0.9938) and 0.0060 to 0.0073 (average = 0.0065), respectively (Table 1). The minimum spanning network showed many unique haplotypes, with only 28 of the 132 haplotypes shared by more than one individual. These unique haplotypes were connected to the center haplotype that occurred in Ishigaki, Shitiping, Checheng, and Badian. The most common haplotype consisted of seven individuals from all locations except Liuqiu and Wanlitong. Closely related haplotypes consisted of individuals from distinct regions, revealing no obvious geographic pattern (Fig. 4).

Figure 4 Minimum spanning network built from 185 concatenated mtDNA sequence (1,336 bp) of U. micropterus with 132 haplotypes.

Colors represent correspondent sampling sites; the size of each pie chart is proportional to the number of individuals; hollow circles are haplotypes that were not collected in this study.

The pairwise ΦST values among sampling sites ranged from −0.025 to 0.121. Liuqiu revealed low but significant genetic variations with all sampling sites except Green Island. Only Liuqiu versus Checheng was statistically significant after the most conservative Bonferroni correction (Table 2). All pairwise ΦST values among groups with different classes of TGC remained low and insignificant (Table S1). The AMOVA results showed that over 99% of variations occurred at the population level under both groupings. Only the ΦSC of different ocean current conditions revealed low but statistically significant structure (ΦSC = 0.017, P < 0.05) (Table 3). The results of genetic analysis support the conclusion that U. micropterus in the study areas should be considered genetically homogeneous with weak genetic structure.

Table 2 Pairwise ΦST values between locations analyzed from the concatenated mtDNA sequence (1,336 bp).

	Japan	Taiwan	Philippines	
	IG	ST	JH	GI	CC	KT	LQ	BD	
IG									
ST	−0.015								
JH	−0.014	−0.006							
GI	0.010	−0.014	−0.016						
CC	−0.009	−0.004	−0.004	−0.005					
KT	−0.009	−0.011	−0.016	−0.009	−0.020				
LQ	0.089*	0.090**	0.044*	0.065	0.116***	0.121**			
BD	−0.025	−0.002	−0.002	0.005	−0.003	−0.004	0.102**		
Notes.

IG Ishigaki

ST Shitiping

JH Jihui

GI Green Island

CC Checheng

KT Wanlitong

LQ Liuqiu

BD Badian

* P < 0.05.

** P < 0.01.

*** P < 0.001.

Bold, significant after Bonferroni correction.

Table 3 AMOVA results for the concatenated mtDNA sequence (1,336 bp) based on two hypothetical groupings.

Source of variations	Degree of freedom	Sum of squares	% of variation	Fixation index	
Three different latitudinal groups: Ishigaki (IG) vs. Taiwan (ST, JH, GI, CC, KT, LQ) vs. Badian (BD)	
Among groups	2	8.4	−0.97	−0.010 (ΦCT)	
Among populations within groups	5	28.3	1.35	0.013 (ΦSC)	
Within populations	177	781.3	99.61	0.004 (ΦST)	
Three different ocean current conditions: (IG, ST, JH, GI) vs. (CC, KT, LQ) vs. (BD)	
Among groups	2	7.7	−1.12	−0.011 (ΦCT)	
Among populations within groups	5	29.1	1.67	0.017* (ΦSC)	
Within populations	177	781.3	99.45	0.006 (ΦST)	
Notes.

IG Ishigaki

ST Shitiping

JH Jihui

GI Green Island

CC Checheng

KT Wanlitong

LQ Liuqiu

BD Badian

* P < 0.05.

Pelagic larval duration and growth rate

The pelagic larval duration represented by the T GC ranged from 33 to 98 days for the eels examined (Table 4). The maximum TGC of 98 days occurred in Liuqiu and the minimum of 33 days occurred in Badian. Liuqiu specimens had the longest and most variable TGC. Ishigaki and Taiwanese specimens had significantly longer TGC than those from Badian (Tukey HSD, P < 0.01). The otolith growth rate from the first feeding ring to the GC represented by GGC ranged from 0.91 to 2.40 µm day −1. The maximum GGC of 2.40 µm day−1 occurred in Badian and the minimum value of 0.91 µm day−1 occurred in Jihui. Significantly lower GGC were observed in Ishigaki and Taiwanese specimens compared with Badian (Tukey HSD, P < 0.01).

Table 4 Detailed data on ranges, means, and statistical analyses of otolith growth rates and increments for all locations used in this study.

Otolith increments calculated from first feeding check (FFC) to growth check (GC) are represented as TGC, and otolith growth rate is represented as GGC.

		Ishigaki	Taiwan	Badian	
		IG	ST	JH	GI	CC	KT	LQ	BD	
TGC	Range	41–68	42–69	39–69	44–74	40–85	41–66	46–98	33–56	
(days)	Mean***	54.6 ± 7.7ab	51.8 ± 7.2b	51.7 ± 7.9b	55.2 ± 9.1ab	57.6 ± 9.7ab	55.2 ± 6.9ab	61.2 ± 14.6a	43.9 ± 4.9c	
GGC	Range	1.04–1.55	0.95–1.58	0.91–1.87	0.99–1.66	0.93–1.73	1.16–1.57	0.98–1.79	1.22–2.40	
(µm day−1)	Mean***	1.28 ± 0.16a	1.36 ± 0.14a	1.39 ± 0.17a	1.32 ± 0.18a	1.31 ± 0.21a	1.33 ± 0.11a	1.38 ± 0.22a	1.60 ± 0.19b	
	N	14	36	31	20	31	11	12	39	
Notes.

IG Ishigaki

ST Shitiping

JH Jihui

GI Green Island

CC Checheng

KT Wanlitong

LQ Liuqiu

BD Badian

*** P < 0.001 (ANOVA test).

Numbers with the same superscript letters (i.e., a, b, ab) are not significantly different (Tukey HSD test, P ≥ 0.05).

The eels were divided into three latitudinal groups according to sampling site, namely Ishigaki (n = 14), Taiwan (Shitiping, Jihui, Green Island, Checheng, Wanlitong, and Liuqiu; n = 141) and Badian (n = 39). The respective mean ± SD daily age and otolith growth rate were 54.6 ± 7.7 days with 1.28 ±0.16 µm day−1, 54.6 ± 9.3 days with 1.35 ± 0.18 µm day−1, and 43.9 ± 4.9 days with 1.60 ±0.19 µm day−1. Ishigaki specimens had the lowest growth rate, whereas the Badian specimens had the highest growth rate and shortest mean TGC (Table 5, Figs. 5A & 6). There were no significant differences in TGC and G GC between the Ishigaki and Taiwanese specimens (Tukey HSD, P >0.05), and both were significantly different from the Badian specimens (Tukey HSD, P < 0.001). The percentage of TGC peaked at 56–60 days (29%) in the Ishigaki specimens, 46–50 days (24%) in the Taiwanese specimens, and 41–45 days (44%) in the Badian specimens (Fig. 5A). U. micropterus tended to have longer TGC and lower GGC at higher latitudes.

Table 5 Otolith increments and growth rates from first feeding check (FFC) to growth check (GC) based on three latitudinal groups.

		Ishigaki	Taiwan	Badian	
		IG	ST, JH, GI, CC, KT, LQ	BD	
TGC	Range	41–68	39–98	33–56	
(days)	Mean***	54.6 ± 7.7a	54.6 ± 9.3a	43.9 ± 4.9b	
GGC	Range	1.04–1.55	0.91–1.87	1.22–2.40	
(µm day−1)	Mean***	1.28 ± 0.16a	1.35 ± 0.18a	1.60 ± 0.19b	
	N	14	141	39	
Notes.

IG Ishigaki

ST Shitiping

JH Jihui

GI Green Island

CC Checheng

KT Wanlitong

LQ Liuqiu

BD Badian

*** P < 0.001 (ANOVA test).

Numbers with the same superscript letters (i.e., a, b) are not significantly different (Tukey HSD test, P ≥ 0.05).

Figure 5 Frequency distributions of PLD based on (A) Ishigaki (n = 14), Taiwan (n = 141), and Badian (n = 39) and (B) Eastern (Shitiping, Jihui, and Green Island, n = 87) and Southwestern Taiwan (Checheng, Wanlitong, and Liuqiu, n = 54).

Figure 6 Sequential changes of increment widths from first feeding check (FFC) to growth check (GC) based on three latitudinal groups.

Ishigaki (n = 13), Taiwan (n = 126), and Badian (n = 37).

The Taiwanese specimens were further divided into eastern (Shitiping, Jihui, and Green Island; n = 87) and southwestern groups (Checheng, Wanlitong, and Liuqiu; n = 54), which had mean ± SD days and otolith growth rates of 52.6 ± 8.0 days (1.36 ± 0.16 µm day−1) and 57.9 ± 10.5 days (1.33 ± 0.19 µm day−1), respectively. The specimens collected in southwestern Taiwan had significantly longer TGC than those collected in eastern Taiwan (Tukey HSD, P < 0.01), whereas there was no significant difference in GGC (Tukey HSD, P > 0.05) (Table 6). The percentage of TGC peaked at 46–50 days (30%) for the eastern Taiwan group and 51–55 days (22%) for the southwestern Taiwan group (Fig. 5B). U. micropterus from southwestern Taiwan had longer TGC.

Table 6 Otolith increments and growth rates from first feeding check (FFC) to growth check (GC).

Specimens from Taiwan are divided into eastern and southwestern groups to test the effect of different current conditions on early life history traits.

		Ishigaki	Southwestern Taiwan	Eastern Taiwan	Badian	
		IG	CC, KT, LQ	ST, JH, GI	BD	
TGC	Range	41–68	40–98	39–74	33–56	
(days)	Mean***	54.6 ± 7.7ab	57.9 ± 10.5a	52.6 ± 8.0b	43.9 ± 4.9c	
GGC	Range	1.04–1.55	0.93–1.79	0.91–1.87	1.22–2.40	
(µm day−1)	Mean***	1.28 ± 0.16a	1.33 ± 0.19a	1.36 ± 0.16a	1.60 ± 0.19b	
	N	14	54	87	39	
Notes.

IG Ishigaki

ST Shitiping

JH Jihui

GI Green Island

CC Checheng

KT Wanlitong

LQ Liuqiu

BD Badian

*** P < 0.001 (ANOVA test).

Numbers with the same superscript letters (i.e., a, b, ab) are not significantly different (Tukey HSD test, P ≥ 0.05).

Discussion

Early life history traits

The diameters of fertilized eggs and total lengths of newly hatched preleptocephalus larvae of U. micropterus were 3.0–3.5 and 10.0 mm (CH Chen, 1994, unpublished data). The total lengths of U. micropterus leptocephali are approximately 60.0 mm at metamorphosis, based on observations of two Uropterygius spp. in the early metamorphosis stage with total lengths of 51.1 and 56.5 mm (Okiyama, 2014). Therefore, the larval growth rates of U. micropterus with PLD of 33–98 days are likely to be between 0.5 and 1.5 mm day−1. This fast larval growth rate greater than 1 mm day−1 was also observed in four other eel species in the eastern Gulf of Mexico (Gymnothorax saxicola, Ophichthus gomesii, Ariosoma balearicum, and Paraconger caudilimbatus) (Bishop, Torres & Crabtree, 2000).

In the present study, the commencement of leptocephalus metamorphosis was defined by the GC where the Sr/Ca ratio drastically decreased and the otolith growth increment widths abruptly increased, similar to findings in other muraenids and eel species (Marui et al., 2001; Correia et al., 2004; Ling, Iizuka & Tzeng, 2005). Although the biological meaning of GC remains unclear, the first growth increment to this check should contain most of the leptocphalus stage and sufficiently represent the PLD of U. micropterus among sampling locations.

Insignificant pairwise ΦST values among groups with different PLDs (Table S1) indicate that the plasticity of the PLD is likely due to individual acclimatization rather than different genotypes. Although PLD variation can be explained by the different growth conditions or birthplaces of the larvae, clearly characteristic PLDs were still observed among the groups in this study, indicating that regional environmental factors may influence tendencies in early life history traits (Searcy & Sponaugle, 2000; Sponaugle & Pinkard, 2004; Bay et al., 2006). Leis et al. (2013) suggested three general classes of factors that might lead to differences in larval dispersal, including biological (e.g., spawning mode and PLD), physical (water movement and habitat fragmentation) and biophysical differences (principally temperature). In this study, U. micropterus at relatively higher latitudes tend to have lower otolith growth rates and longer PLDs compared with specimens at lower latitudes, potentially owing to water temperature. The sea surface temperatures (SSTs) during the spawning season and pelagic leptocephalus stage of U. micropterus (Chen, 1997) were derived from Global Sea Temperature (http://www.seatemperature.org/) (Fig. S1). The SST of Itoman (Japan) is 5 °C lower than that of Guihulngan (the Philippines) in May and December, indicating that the leptocephali in the different study areas experienced different SSTs regardless of whether the eels were self-recruited or transported from other habitats. The effects of temperature on larval development were discussed in rearing experiments that demonstrated positive correlations in Anguilla japonica glass eels and elvers (Fukuda et al., 2009). Leptocephali at lower latitudes with higher SSTs may grow faster and require less time to reach the minimum size for metamorphosis (Reveillac et al., 2008). Because the latitude of Ishigaki is similar to that of eastern Taiwan, neither the mean PLDs nor the otolith growth rates were expected to be different.

Since the effect of SST on larval growth can be excluded from the sampling sites in Taiwanese waters, the relatively longer PLD in the southwestern specimens might be attributable to different current systems or recruitment routes. Larval dispersal route is considered to be strongly influenced by ocean currents (Kim et al., 2007). For instance, newly hatched larvae of C. myriaster can be retained in mesoscale eddies for several months, resulting in longer PLD than expected (Kurogi et al., 2012). The strong Kuroshio Current flows northward along the east coast of Taiwan year-round (Rudnick et al., 2011). When leptocephali competent to metamorphose drift to this area, they can instantly metamorphose and settle. However, the current system in southwestern Taiwan is affected by different water masses seasonally, including the intrusion of the Kuroshio branch, the substitution of South China Sea surface water mass into the Taiwan Strait, and other transient oceanographic events (Shaw, 1991; Farris & Wimbush, 1996; Hu et al., 2010). In addition, a weak anticyclonic eddy with a diameter of 100–200 km was identified in this area (Fig. 1, Fig. S2). The leptocephali might entrain in the anticyclonic eddy or complicated current system and therefore require a longer time to settle along the southwest coast of Taiwan.

Population structure and larval dispersal

Previous studies have shown that the dispersal or retention of larvae may greatly influence gene flow for demersal fishes (Taylor & Hellberg, 2003). During the field collection stage, we found that U. micropterus inhabited shallow water close to the upper tidal zone, which was usually shallower than 1 m and was exposed to air during low tide. These habitats are usually fragmented, but each has an extremely high density of individuals, suggesting that some larvae hatched in the upper tidal zone may remain in nearshore areas and contribute to local habitats. However, the fast-flowing Kuroshio Current is frequently used by the larvae of many eel species for transportation to a wide range of areas (Miller et al., 2002), which may facilitate U. micropterus gene flow. The sampling site at Badian is relatively isolated from the main oceanic current systems around the Philippine archipelago. Compared with the Taiwanese sites in this study, there may be fewer U. micropterus leptocephali drifting away from Badian per generation. It is likely that a handful of migrants have sufficiently contributed to the genetic homogenization among the study regions over timescales of tens to hundreds of thousands of years (Reece et al., 2010).

Nevertheless, it seems that limited larval exchange over time is inadequate to explain the significant genetic divergence of Liuqiu from all locations but Green Island with Φ ST before Bonferroni correction. Unknown environmental mechanisms that restrict larval dispersal and enhance massive self-recruitment may be a possible explanation. Previous studies have suggested that the self-recruitment of larvae is a common phenomenon for marine fishes and invertebrates, disregarding larval dispersal capabilities (Taylor & Hellberg, 2003; Teske et al., 2015). Self-recruitment of leptocephali is also surmised more often than dispersal in some areas and is attributed to local current systems, semi-enclosed ocean environments, or the swimming ability of the larvae (Miller et al., 2011; Miller et al., 2016). Furthermore, the PLD of Liuqiu specimens was the longest and most variable (46 to 98 days) of all sampling sites, indicating that U. micropterus larvae had more complicated composition and different transportation routes to Liuqiu. The longer PLD implies that some larvae might recruit from places with cooler water or remote populations not included in this study. Further studies with larger-scale sampling among a range of habitats may provide more details on the population genetic structure of U. micropterus.

Conclusion

In the present study, intraspecific variations in PLD were found in U. micropterus among defined groups without obvious population genetic structure. These variations were likely acclimatization-dependent rather than genotype-dependent. Weak divergence of U. micropterus was observed in Liuqiu, southwestern Taiwan, most likely owing to the different recruiting routes of the leptocephali. This study suggests that the intraspecific variation in the PLDs of the eels might have resulted from different seawater temperatures and complex ocean conditions.

Supplemental Information

Table S1 Pairwise ΦST values (below diagonal) and P values (above diagonal) between different PLD groups analyzed from the concatenated mtDNA sequence (1,336 bp)

Click here for additional data file.

Figure S1 Sea surface temperature of three locations, which can represent three latitudinal groups from May to December

The time period that includes the spawning season and pelagic leptocephalus stage of Uropterygius micropterus. Data were obtained from the Global Sea Temperature website (http://www.seatemperature.org/).

Click here for additional data file.

Figure S2 Map of currents around Taiwan plotted by an acoustic Doppler current profiler (ADCP) using sea surface water (¡50 m depth) data from May to December in 1991–2012

Data were obtained from the Ocean Data Bank of the Ministry of Science and Technology, Republic of China (http://www.odb.ntu.edu.tw/).

Click here for additional data file.

Supplemental Information 1 DNA sequencing (MF190188 to MF190364)

The corresponding accession numbers are in another text file.

Click here for additional data file.

Supplemental Information 2 DNA sequencing (MF190188 to MF190364) (text)

Click here for additional data file.

We are grateful to Dr. Yuichi Kano, Florence Evacitas, Jiunn-Shiun Chiou, Ming-Tai Chou, Ying-Ching Yuan, Li-Chi Cheng, Yun-Ting Hsieh, Zen-Wei Lin, Kai-Chiang Chang, Po-Wei Huang, Yu-Wei Liu, Chu-Feng Lu and Chang-Chun Liu for their assistance in specimen collection; Dr. Hong-Ming Chen for providing reproductive information on the eels; and Sih-Yu Chen, Han-Ting Huang, and Jhen Hsu for assistance in map drawing. We specially thank Joel Corush and Margaret Scholten for assistance in English editing.

Additional Information and Declarations

Competing Interests

Author Contributions

Animal Ethics

Data Availability

The authors declare there are no competing interests.

Wen-Chien Huang conceived and designed the experiments, performed the experiments, analyzed the data, prepared figures and/or tables, authored or reviewed drafts of the paper, approved the final draft.

Jui-Tsung Chang conceived and designed the experiments, authored or reviewed drafts of the paper, approved the final draft.

Chun Liao conceived and designed the experiments, performed the experiments, authored or reviewed drafts of the paper, approved the final draft.

Atsushi Tawa performed the experiments, contributed reagents/materials/analysis tools, authored or reviewed drafts of the paper, approved the final draft.

Yoshiyuki Iizuka analyzed the data, contributed reagents/materials/analysis tools, authored or reviewed drafts of the paper, approved the final draft.

Te-Yu Liao and Jen-Chieh Shiao conceived and designed the experiments, contributed reagents/materials/analysis tools, prepared figures and/or tables, authored or reviewed drafts of the paper, approved the final draft.

The following information was supplied relating to ethical approvals (i.e., approving body and any reference numbers):

The experimental species is a non-regulated animal, and there is no relevant provision for ethical review process in Taiwan.

The following information was supplied regarding data availability:

The COI and Cytb sequences described here are deposited in GenBank with accession numbers to MF190188 to MF190364.

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
