# Peer review of "Pelagic larval duration, growth rate, and population genetic structure of the tidepool snake moray Uropterygius micropterus around the southern Ryukyu Islands, Taiwan, and the central Philippines"

_PeerJ, doi:10.7717/peerj.4741_

## Round 0.1 · original submission · Minor Revisions

This is a useful paper that should be published. However, the English does need editing by a native english speaker. Each of the three reviewers made useful points that you need to attend to in your revision, and indicated that there are no serious issues. The Leis et al 2013 review of latitudinal variation in dispersal ( DOI: 10.1098/rspb.2013.0327) is a particularly important reference to incorporate in your discussion.

Reviewer 1 ·

Basic reporting

needs a rewriting by a fluent English speaker.. many phrases are imprecise and vague and not standard usage and, as well, there are glaring errors with dropping of plurals- like "various PLD(s) and growth rate(s) of" & dropping of indefinite and definite articles and prepositions, like "(The) population structure of fishes", "(a) handful (of) congrids" & "breed at (a) (single?) specific spawning area".
Many references, esp. for larvae are old and/or referring to gobies- completely different fish models.
discuss Leis et al 2013 review (Does fish larval dispersal differ between high and low latitudes?)

Experimental design

the data are fine, it is the presentation and discussion that are below standard

Validity of the findings

discussion is inadequate.. some statements are unsupported, some citations do not match the phrase they are being referenced to.. the authors need to carefully go over the wording of the discussion to limit it to what exactly they know based on what from the literature.

Additional comments

specific comments by line
> 38 several means less than many.. and there are many factors!
> 42 PLD is not used to predict population structure... use site-attached instead of benthic, which means sitting on the bottom.
> 44 PLD often "correlates", just not very well.. how about "PLD has been found to strongly influence population structure only in..."
> 51 "distinct early life history traits, leading to specific PLDs" I have no idea what this means, vague terms, everything on earth has "distinct traits".
> 53 "remain unclear and barely evaluated".. some people's entire careers evaluate this! so not barely!
> 57 the unusual leptocephalus is not important at all.. it is the pelagic larval stage that is meaningful.. the actual length of the PLD.. for example, many labrids have the same PLDs and the same lack of genetic structure- the actual shape of the larva is irrelevant.
> 60 & 62.. English grammar, need to use "have been" instead of "were" for things that happen continuously in the recent past...
> 63 varying PLDs
> 68 "genetic structure among populations, geographic areas or environmental systems" is vague.. populations are in different areas, and "environmental systems" is a meaningless word.
> 69 use "explored thoroughly".. "discussed" implies it was studied well but not discussed well.
> 71 Anguilliformes
> 86 fish have sperm, not semen
297: with a long PLD and lepto stage, being spawned in the intertidal would not be expected at all to cause retention.. maybe for fish that brood large eggs in the intertidal, maybe, but never eels.
299- no despite necessary- they are long PLD larvae.. "distinct" PLD is meaningless.
303- there must be some dispersers.. these statements are baseless..
307- sentence is incomprehensible- ecological explanations irrelevant to long distance transport- dispersal occurs with the infrequent offshore dispersal, not any retained larvae.. genetic homogeneity can be maintained by a single disperser every 100 years..
312: "Self-recruitment(s) of leptocephali are also observed more often than dispersal in".. they cannot have shown that- it would be impossible to study how many dispersed..
317- "longer PLD implies that some larvae might recruit from remote populations".. but cooler water means slower growth and/or longer PLD, so can never separate those two causes.

·

Basic reporting

Clear, unambiguous. Literature references are sufficient. Structure of article is sound, data are presented relative to specific questions/hypotheses.

Experimental design

Question is well-defined and relevant. Investigation performed with high technical savvy and method provide sufficient detail to replicate.

Validity of the findings

Sound, well executed. Interpretation of data is meaningful and clear.

Additional comments

This is a study that includes 195 specimens analyzed for 1,336 nucleotides of concatenated cyt b and coi sequences from one species of moray eel surveyed in SE Asia. Their goal was to assess PLD and genetic structure. They find that eels with longer PLD and lower otolith growth rate were at higher latitudes with colder water; i.e. faster growth and shorter pelagic periods in warmer temperatures. They find low genetic structure at this scale, which is not surprising given previous phylogeographic studies of morays. Generally, I find the study to be interesting and very well executed. With some minor mistakes, the manuscript is well written, the figures are clear and interpretable, the data is presented fairly plainly, and the analyses and interpretations seem very sound. I have no problem seeing this published as it stands or with very minor revision.

The only criticism I can provide is just a suggestion that the premise or setup could use some finessing. There is a known, but shaky, relationship between PLD and genetic structure. PLD is known to be high in morays and genetic structure is known to be low or nonexistent. The authors find that PLD is indeed high and genetic structure is very low. The paper is better presented as a descriptive piece. It's a straw man to present the finding of low genetic structure as anything but expected. The interesting stuff here is yet another moray with empirical estimates of PLD, the finding of differential PLD across latitudes even in this fairly modest spatial scale, and the repeated finding of low genetic structure in another moray. The introduction sets this up a bit as if it is a question whether or not PLD will be long and genetic structure low - whereas that finding is all but expected.

Minor edits:
" Results: no missing data? Please clarify extent of any missing markers from concatenated dataset
" 292: title, larval not "lraval"
" 294: change "the gene flows" to "gene flow"

·

Basic reporting

This study is an assessment of links between population-level differences in pelagic larval duration (PLD) and population genetic structure. This is an innovative idea that has not been tested previously to my knowledge.

There were a number of minor typographical errors (see below) but these are readily corrected in a few minutes. Overall the English was very good, and the result is an efficient, highly-readable report. Introduction is informative and succinct, literature is well covered, overall structure is appropriate. With one or two exceptions, see below, figures are clear and justified.

Experimental design

The research is within the scope of PeerJ and the study is well-designed. Statistical analyses are robust, using the very conservative Bonferroni correction. The only criticism that could be leveled about study design is that the tidepool snake moray occurs from Africa to Micronesia, so that the geographic scale of sampling could be expanded profitably. However, the authors know this, and I won’t criticize them for lacking unlimited funding and resources.

Validity of the findings

The results include significant differences in PLD along a latitude (or temperature) gradient. Authors conclude that differences in PLD between sample sites are not due to population genetic isolation, but due to differences in environment. This conclusion seems robust.

The only significant phi-st between sites was with Liuqiu versus Checheng, closely situated sites in southern Taiwan. Liuqiu also had the longest and most variable PLD. Authors address this in Discussion, invoking possible long distance recruitment within the South China Sea. Perhaps add a sentence at the beginning of this discussion that Liuqiu is significantly different from almost every location with phi-st prior to Bonferroni correction (which is highly conservative).

Additional comments

138 Call-outs for Figures 5 & 6 are out of order. Maybe eliminate the call-outs as they aren’t needed here.

228-229 and elsewhere: P values should have same number of significant figures.

Table 1 should have a horizontal line between Japan and Taiwan.
Table 3 could go in supplemental information, a bunch of nonsignificant phi-st values.
Figure 7 could also go in supplemental information, as it is peripheral to main theme.

Rewording Suggestions
32 ‘Gene flow’ should be singular
33 Perhaps replace ‘poor’ with ‘weak’.
51 Replace ‘undergo’ with ‘experience’.
56 Delete ‘handful’.
62 “There are only a few studies…’.
65 ‘Areas’ should be plural.
95 ‘Zones’ should be plural.
100, 171 Numerous
126 Replace ‘have been indicated’ with ‘are typically’.
133 replace the first ‘in’ with ‘the’.
163 ‘pairwise’ should be lower case.
204 Sampling
211 ‘Structure’ should be singular.
245 I think this should be ‘metamorphosis’.
249 This should read ‘four other eel species…’
256 This should read ‘…to this check should contain most of the leptocphalus stage…’
287 This should read ‘…other transient oceanographic events…’
302 ‘gene flow’ should be singular.
304 this should read ‘…effectively a handful of migrants…’
306 ‘…timescales of tens…’
309 ‘…maybe a possible…’

Overall, an innovative study design that should be emulated in other fishes, to test the generality of findings. This was an informative, enjoyable read that should appeal to the broad readership of PeerJ. Congratulations to authors.

Possibly Useful Reference:
Selkoe KA, Gaggiotti OE, ToBo Lab, Bowen BW, Toonen RJ. 2014 Emergent patterns of population genetic structure for a coral reef community. Mol. Ecol. 23, 3064–3079. (doi:10.1111/mec.12804)

---

## Round 0.2 · accepted · Accept

This all looks ok now...you had three good reviews and have responded appropriately.

#